# The Protein Kinase Receptor Modulates the Innate Immune Response against Tacaribe Virus

**DOI:** 10.3390/v13071313

**Published:** 2021-07-07

**Authors:** Hector Moreno, Stefan Kunz

**Affiliations:** Institute of Microbiology, Lausanne University Hospital, CH-1011 Lausanne, Switzerland

**Keywords:** protein kinase receptor (PKR), mammarenavirus, interferon, innate immune response, Mx1, ISG15, CCL5

## Abstract

The New World (NW) mammarenavirus group includes several zoonotic highly pathogenic viruses, such as Junin (JUNV) or Machupo (MACV). Contrary to the Old World mammarenavirus group, these viruses are not able to completely suppress the innate immune response and trigger a robust interferon (IFN)-I response via retinoic acid-inducible gene I (RIG-I). Nevertheless, pathogenic NW mammarenaviruses trigger a weaker IFN response than their nonpathogenic relatives do. RIG-I activation leads to upregulation of a plethora of IFN-stimulated genes (ISGs), which exert a characteristic antiviral effect either as lone effectors, or resulting from the combination with other ISGs or cellular factors. The dsRNA sensor protein kinase receptor (PKR) is an ISG that plays a pivotal role in the control of the mammarenavirus infection. In addition to its well-known protein synthesis inhibition, PKR further modulates the overall IFN-I response against different viruses, including mammarenaviruses. For this study, we employed Tacaribe virus (TCRV), the closest relative of the human pathogenic JUNV. Our findings indicate that PKR does not only increase IFN-I expression against TCRV infection, but also affects the kinetic expression and the extent of induction of Mx1 and ISG15 at both levels, mRNA and protein expression. Moreover, TCRV fails to suppress the effect of activated PKR, resulting in the inhibition of a viral titer. Here, we provide original evidence of the specific immunomodulatory role of PKR over selected ISGs, altering the dynamic of the innate immune response course against TCRV. The mechanisms for innate immune evasion are key for the emergence and adaptation of human pathogenic arenaviruses, and highly pathogenic mammarenaviruses, such as JUNV or MACV, trigger a weaker IFN response than nonpathogenic mammarenaviruses. Within the innate immune response context, PKR plays an important role in sensing and restricting the infection of TCRV virus. Although the mechanism of PKR for protein synthesis inhibition is well described, its immunomodulatory role is less understood. Our present findings further characterize the innate immune response in the absence of PKR, unveiling the role of PKR in defining the ISG profile after viral infection. Moreover, TCRV fails to suppress activated PKR, resulting in viral progeny production inhibition.

## 1. Introduction

Mammarenaviruses are a large genus of viruses divided into Old and New World arenavirus groups (OW and NW, respectively), according to antigenic properties, phylogeny, and geographic distribution [1]. Both groups include zoonotic viruses that are highly pathogenic to humans, such as Lassa (LASV), JUNV, MACV, and Guaranito virus [2]. The prototypic OW arenavirus Lymphocytic Choriominingitis virus (LCMV) is a neglected pathogen with world-wide distribution and clinical significance in immunocompromised individuals and pregnant women [3]. The highly diverse NW arenavirus group is further divided into four clades: A, B, C, and D. While several members of clade B are confirmed human pathogens, some clade D viruses also show potential for viral emergence [4,5]. Mammarenaviruses are enveloped, bi-segmented, negative-stranded viruses with a life cycle restricted to the cytosol [6]. The small genomic segment encodes the glycoprotein precursor and the viral nucleoprotein (NP), while the large segment codes for the matrix protein (Z) and the RNA-dependent RNA-polymerase (L). NP and L, together with the cis-acting sequences of the viral genome, are necessary for the virus ribonucleoprotein (RNP) complex formation, which is needed for the replication and transcription processes [7,8].

Mammarenavirus infection is typically detected by RIG-I-like receptors and Toll-like receptors. The activation of these pattern recognition receptors (PRRs) triggers an IFN-I immune response, upregulating a plethora of ISGs, which encode effector proteins that have an antiviral effect and induce a cellular antiviral status. Mammarenaviruses are capable of inhibiting the IFN-I response to different extents, via a 3′–5′ exoribonuclease NP domain [9,10,11,12], and highly pathogenic arenaviruses trigger a weaker induction of the IFN-I response than their nonpathogenic counterparts [12,13,14,15,16], suggesting that the capacity to overcome or suppress the IFN-I response is relevant for causing disease in humans. Indeed, the attenuated JUNV Candid#1 triggers a stronger IFN-β expression than the highly pathogenic Romero strain [17]. Moreover, the Z protein of pathogenic mammarenaviruses was shown to inhibit RIG-I, essential for the IFN-I response [16], summing to the immunosuppressive activity of NP. Despite the inhibitory effect of NP and Z over IFN-I response, and contrary to OW arenaviruses, NW arenaviruses fail to completely abolish IFN-I response in human cells [10,12,14,15,18].

In addition to the aforementioned host’s PRRs, recent studies from others and us revealed that the dsRNA-PKR plays an important role during NW arenavirus infection [14,18,19]. PKR is an ISG that contributes to the enhancement of the IFN-I response against measles virus [20,21], West-Nile virus [22], or upon IFN-I treatment [23]. Upon detection and recognition of foreign dsRNA, PKR undergoes autophosphorylation and subsequently phosphorylates the α subunit of eIF2α, leading to the inhibition of protein cap-dependent translation [24]. Interestingly, a recent study showed that, in contrast to LASV, highly pathogenic NW arenaviruses accumulate dsRNA during infections [25], possibly leading to the observed colocalization of the viral RNP with RIG-I, the melanoma differentiation-associated protein 5 (MDA5), and phosphorylated PKR [19,26]. Moreover, it has been shown that the highly pathogenic arenaviruses, JUNV Romero Strain and MACV, but not LASV, induce higher IFN-β levels in PKR null cells than in non-transduced parental cells [18]. Nevertheless, previous results in our lab showed that PKR partially controls the infection by the nonhuman pathogenic TCRV, but had no impact on the infection by the attenuated JUNV Candid#1 strain [14].

Altogether, the literature suggests that the role of PKR in the IFN-I response might differ among pathogenic, attenuated, or nonpathogenic mammarenavirus infection. In the present study, we investigated and characterized the innate immune response triggered by nonpathogenic arenavirus TCRV in PKR KO cells, compared to parental cells subjected to an analogous mock clustered, regularly interspaced, short palindromic repeat–associated 9 (CRISPR/Cas9) engineering. Our results indicated that PKR changes the expression kinetic of Mx1 and ISG15, but not CCL5, whose expression is inhibited, maintaining a comparable pattern in presence and absence of PKR. Moreover, our findings indicated that activated PKR controls TCRV infection, inhibiting the viral progeny production.

## 2. Materials and Methods

### 2.1. Cells, Viruses and Infections

Scrambled A549 control (A549/Scr) and PKR KO A549 (A549/PKR KO) cells were obtained as described in [14]. Briefly, all cells were maintained in Dulbecco’s modified Eagle medium with high glucose (4.5 mg/liter) and GlutaMAX (DMEM; Gibco BRL) with 10% (vol/vol) fetal calf serum (FCS) and held in an CO_2_ incubator (37 °C and 5% (vol/vol) CO_2_). TCRV (strain 11573) was plaque purified and propagated in VeroE6 and baby hamster kidney (BHK) cells, followed by PEG-precipitation and sucrose cushion purification, as described in [14].

For infections, cells were seeded 48 h in advance and counted before infection. The multiplicity of infection (MOI) was determined as described in each particular experiment. Before each experiment, cells were tested for mycoplasma contamination using a MycoAlert mycoplasma detection kit (Lonza, Basel, Switzerland). Inoculums were prepared by diluting the desired amount of virus in DMEM/10% FCS and incubated with cells for 90 min in a CO_2_ incubator. Upon adsorption, the inoculums were removed and fresh DMEM/10%FCS was added to each well. For infections in rIFN-αA/D-stimulated cells, 24 h after seeding, cells were treated with 100 U/mL for 24 h. Cells were then infected as described above.

### 2.2. Antibodies and Reagents

Anti TCRV NP MA03-BE06 antibody [27] was obtained from BEI Resources (Manassas, VA, USA). Rabbit antibody against ISG15 was obtained from Cell Signaling Technology (Danvers, MA, USA). Rabbit antibody against Mx1 was purchased from Proteintech (Rosemont, IL, USA). Goat antibody against CCL5 was purchased from R&D Systems (Minneapolis, MN, USA). Antibody against Vinculin (EPR8185) was obtained from Abcam (Cambridge, UK). Recombinant human IFN (interferon-αA/D human; #I4401) was purchased from Sigma-Aldrich (St. Louis, MO, USA). Alexa Fluor-488 F(ab′)2 fragment of goat anti-mouse IgG was obtained from Life Technologies (Carlsbad, CA, USA). Polyclonal rabbit anti-mouse and donkey anti-goat antibody conjugated with horseradish peroxidase (HRP) were obtained from Dako (Santa Clara, CA, USA).

### 2.3. Immunofocus Assay (IFA)

For viral titer quantitation by IFA, supernatants were cleared from cellular debris by centrifugation at 1200 rpm for 3 min, and stored at −80 °C until analysis. Samples were 10-fold serially diluted in DMEM/10% FCS and used to infect previously prepared VeroE6 cells in 96-well plate format. After 16–20 h of infection, cells were washed with PBS, and fixed with 2% (wt/vol) formaldehyde for 30 min at room temperature. Then, cells were permeabilized with PBS/0.1% saponin/1% FCS (working solution) for 30 min at room temperature. MA03-BE06 antibody was used as the primary antibody to detect TCRV-NP, diluted 1:500 in working solution and applied to the cells for 1 h at room temperature in a rocking station. Alexa 488-congugated anti-mouse IgG1 was used as the secondary antibody, diluted 1:500 in working solution, and applied to cells for 45 min at room temperature in a rocking station. Before scoring the samples, cells were washed three times with PBS. Positive infectious foci were scored using an EVOS FLoid cell imaging station with a 20× Plan Fluorite Lens (Thermo Fisher Scientific).

### 2.4. RNA Extraction, RT, qPCR and RT2 Profiler

Samples collected for RNA extraction were kept in RNAlater (Sigma-Aldrich) at −20 °C until analysis. RNA for IFN-β quantitation was extracted with a NucleoSpin RNA kit (Macherey-Nagel, Düren, Germany) and eluted in 60  microliters of nuclease-free water, in accordance with the manufacturer’s instructions. RNA for RT2 profiler assay was extracted with TRIzol reagent (Invitrogen), following the manufacturer’s instructions and resuspending the RNA pellet in 20 microliters of nuclease-free water. Total RNA quantitation was performed with a Qubit 4.0 instrument (Thermo Fisher Scientific). A total of 0.5 micrograms of total RNA was used for reverse transcription reactions using a high-capacity cDNA reverse transcription kit (Applied Biosystems, Foster City, CA, USA), following the manufacturer’s instructions. TaqMan probes specific targeting human IFN-β (Hs01077958_s1/FAM) and glyceraldehyde-3-phosphate dehydrogenase (GAPDH; Hs99999905_m1/VIC) were obtained from Applied Biosystems. Quantitative PCR (qPCR) was performed using a StepOne qPCR system (Applied Biosystems, Foster City, CA, USA), and relative gene expression to GAPDH was determined following the 2^−^^ΔΔ^Ct calculation (where Ct is threshold cycle). For transcriptome profiling, human antiviral response RT2 Profiler PCR array kits were used (PAHS-122ZG; Qiagen, Hilden, Germany). A total of 3.5 micrograms (quantified by Qubit 4.0) of total cellular RNA was used for the reverse transcription reaction (RT2 SYBR green qPCR mastermix; Qiagen, Hilden, Germany). RT2 Profiler PCR array 384-well plates were set up by a PIRO personal pipetting robot (Labgene, Châtel-Saint-Denis, Switzerland). All RNA samples were tested for quality and integrity in a fragment analyzer (Agilent Technologies, Santa Clara, CA, USA), selecting only those with an RNA quality score (RIN) of 10. All tested samples were tested negative for genomic DNA contamination. Samples with aberrant amplification curves or shifted or multiple melting peaks were discarded from the analysis. qPCR reaction was performed using a LightCycler 480 instrument II (Roche, Basel, Switzerland) in accordance with the manufacturer’s instructions, and relative gene expression to GAPDH was determined following the 2^−^^ΔΔCt^ calculation (where Ct is threshold cycle).

### 2.5. Immunoblotting

Samples were collected and lysed in CelLytic by incubating on ice for 30 min. Then, samples were centrifuged for 15 min at 4 °C at 14.000 rpm, the supernatants were moved to a new clean tube, and stored at −80 °C until analysis. Samples were then loaded with Laemli buffer in Novex Value 4–20% Tris-Glycine precasted gels, and transferred to a nitrocellulose membrane. Membranes were stained with indicated antibodies diluted in 3% powder milk in PBS 0.1% Tween 20. Signals were acquired by an ImageQuant LAS 4000 Mini (GE Healthcare Lifesciences), and quantitation and analyses of Western blot results were performed with ImageJ software.

### 2.6. IFN Quantitation Bioassay

To measure the antiviral activity of IFN produced by A549/Scr and A549/PKR KO cells infected with TCRV, we performed an IFN-I bioassay, as described previously [12,14]. Briefly, VeroE6 were treated cells with UV-inactivated tissue culture supernatants from TCRV-infected A549/Scr and A549/PKR KO cells. As a control, we used titrated amounts of IFN-IIa (0, 10, 100, 1000, and 10,000 IU/mL) and subjected them to the same UV inactivation protocol as the tested samples. UV treatment was performed at 4 °C, in a rocking station at 10 cm from the UV irradiation source for 2 min. After 16 h of incubation, cells were infected with VSV, which is known to be highly susceptible to IFN [28]. After 8 h of infection, cells were fixed and assayed in IFA with specific antibodies against the VSV M protein, as reported previously [14]. The level of infection of VSV is therefore inversely proportional to the IFN produced.

## 3. Results

To investigate the role of PKR in the IFN-I response triggered by a NW arenavirus infection, we measured the levels of IFN-β mRNA in control and PKR knockout human lung epithelial A549 cells (A549/PKR KO) infected with TCRV and JUNV-Candid#1. A549 cells have been extensively used by others and us to recapitulate mammarenavirus infection and represent a reliable model of study (12, 14, 17, 19). To obtain the A549/PKR KO cells, we used CRISPR/Cas9 guide RNAs [29,30], and A549 cells subjected to analogous CRISPR/Cas9 editing with a scrambled guide RNA sequence (A549/Scr) as control cells. First, we infected A549/Scr and A549/PKR KO cells with TCRV at low MOI (0.01 PFU/cell) and collected total cellular RNA after 5 days of infection. As previously shown in [14], at this time after infection, the number of TCRV-infected cells in A549/Scr and A549/PKR KO cells are comparable (93.6 +/− 0.5% and 95 +/− 0.8%, respectively), but the viral titers were significantly higher in the absence of PKR. We observed lower levels of IFN-β transcripts in infected A549/PKR KO cells compared to infected A549/Scr control cells (Figure 1A). To faithfully address the biological consequences the IFN produced upon TCRV infection in A549/Scr and A549/PKR KO cells, we evaluated the antiviral effect of UV-inactivated supernatants from infected cells. To this aim, and as previously described [12,14], we used vesicular stomatitis virus (VSV) as a surrogate to quantify the amount of IFN produced (Figure 1B). These results were in agreement with previous studies on the role of PKR in the innate immune response against different viruses [20,21,22,31], as well as with the previously reported reduced IFN-I response against TCRV [10,12,14].

Next, we further characterized the IFN-I response in PKR null cells by studying the expression profile of ISGs. We compared the expression of 66 ISGs in A549/Scr and in A549/PKR KO cells infected with TCRV. To this aim, we performed infections at low MOI (0.01 PFU/cell) and collected cellular total RNA after 5 days of infection. Coherently with previous results (Figure 1), we observed that depletion of PKR in A549 cells reduces IFN-β expression. The differences in the amplitude of IFN-β expression between single IFN-β RNA quantitation (Figure 1) and in the screening array (Table 1) are likely due to the use of different qPCR methods, as well as the employ of multiple housekeeping genes in the array, which renders a more accurate relative quantitation. Indeed, our results in the bioassay experiment (Figure 1B) and the accurate mRNA quantitation using multiple housekeeping genes are comparable. Interestingly, despite lower IFN-β mRNA levels, Mx1 and ISG15 increased their expression in A549/PKR KO cells (Table 1).

Given the relevance of these effector ISGs in the IFN-I response, we monitored the protein levels of Mx1, ISG15, and CCL5 after TCRV infection. Our results show that the lack of PKR causes a delayed production of Mx1 and ISG15. However, concomitant with the higher gene expression observed (Table 1), Mx1 and ISG15 reach higher protein levels in A549/PKR KO cells than in A549/Scr control cells at late times after infection (Figure 2A). Moreover, the absence of PKR resulted in the overall inhibition of CCL5 expression, without causing any delay in the expression kinetic after TCRV infection (Figure 2B). In line with previous findings [14], we further confirmed that the absence of PKR led to a limited increase of TCRV NP expression during early times postinfection (Figure 2C). The effects of lacking PKR on the levels and dynamics expression of ISGs confirmed the pivotal role of PKR in the innate immune response triggered by TCRV infection.

TCRV NP expression is partially limited by PKR during early times after infection (Figure 2C and [14]), and the results above strongly suggest that PKR activation efficiently controls TCRV infection course via ISG activation. To better elucidate the consequences of activation of PKR during TCRV infection, we stimulated A549/Scr and A549/PKR KO cells with 100 IU/mL of rIFN-αA/D and then infected at MOI of 0.01 PFU/cell. Our results showed that TCRV reached significantly higher viral titers in the absence of activated PKR than in its presence during early times after infection (<3 days postinfection), albeit the production of viral progeny became comparable at later time points (Figure 3). These results indicated that, once activated, PKR partially restricted TCRV growth.

## 4. Discussion

In the present study, we show that, despite causing a reduced expression of IFN-β (Figure 1), depletion of the dsRNA sensor PKR leads to changes in the levels and dynamics of production of ISGs, including Mx1 and ISG15 (Table 1, Figure 2). Previous reports already described PKR as an enhancer of the IFN-I response upon viral infection [20,21,22,31]. Interestingly, in the case of JUNV Candid#1, despite the differences in IFN-β mRNA levels in infected A549/Scr and A549/PKR KO cells, viral progeny production was not affected [14], suggesting that either the contribution of PKR against JUNV infection is not biologically relevant or that JUNV can deploy molecular mechanisms to overcome the host’s innate immune response. Interestingly, a previous study showed that infections with the highly pathogenic JUNV Romero strain and MACV at high MOI (3 PFU/cell) in nontransduced A549 cells and A549/PKR KO cells resulted in an increase of the IFN-β expression [18], suggesting that attenuated and pathogenic JUNV strains may interact with PKR in different manners or efficiencies.

We previously described that TCRV infection is increased in A549/PKR KO cells at late time points, concomitantly with PKR activation in A549 cells [14]. Moreover, we found that similar to JUNV Candid#1, TCRV NP expression is subject to a very limited control of PKR during early stages of the infection (Figure 2) and [19], suggesting that PKR may also be activated during initial virus propagation. In this scenario, PKR could inhibit the early local viral propagation and viral protein synthesis, remaining undetectable by analysis of whole cell lysate, as only a small fraction of PKR is activated. In the absence of PKR, TCRV reaches higher viral titers only at late times postinfection [14], but this effect was observed at earlier times after infection when cells were prestimulated with IFN (Figure 3). These results suggest that a PKR antiviral effect may be effective only for a limited time after its activation. Furthermore, this observation suggests that, despite being able to affect initial viral protein production, the impact on viral progeny production can only occur with extensive PKR activation, either via induced IFN stimulation or via intrinsic viral detection.

Despite rendering lower IFN-β levels, TCRV infection in A549/PKR KO cells results in a delay of Mx1 and ISG15 expression, but also in increased mRNA and protein expression of both host factors at late time points (Table 1 and Figure 2). Importantly, reduced ISG15 and Mx1 expression occurs concomitantly with increased TCRV NP expression (Figure 2 and [14]), suggesting that the observed antiviral effect of PKR may be implemented with the participation of these ISGs. In contrast, CCL5 expression is reduced due to the absence of PKR at both gene and protein levels, and renders a comparable kinetic of expression. Therefore, PKR ablation seems to alter the ISG profile kinetic of Mx1 and ISG15, but not other ISGs, such as CCL5. Our results then suggest that the observed differences in viral progeny production in A549/PKR KO cells infected with TCRV but not with JUNV [14] might be the result of not only reduced IFN-β levels but also of changes in the expression pattern of specific ISGs and altered innate immune response.

CCL5 is a chemokine expressed in many cell types in response to viral infections and IFN-β, and plays a pivotal role in migration of effector and memory T cells [32,33]. CCL5 is a relevant player in the response against arenaviruses, and its absence in mice leads to the establishment of chronic infections of LCMV clone 13 [34]. Moreover, infection with the nonhuman pathogenic NW mammarenavirus Pichinde virus (PICV) p2 strain, which causes mild disease in guinea pigs, was followed by increased CCL5 expression at late time points, in contrast to the virulent PICV p18 strain [35]. The CCL5 reduction due to the lack of PKR may then contribute to worse disease outcome in mammarenavirus infections. In such scenarios, the weaker PKR activation observed in highly pathogenic NW arenaviruses would contribute to a lower CCL5 expression and increased virulence.

ISG15 is strongly induced by the IFN-I cascade and exerts its antiviral activity by being incorporated to nascent peptides in a process, similar to ubiquitination, termed ISGylation, which affects their stability [36]. The expression of ISG15 increases in response to many viral infections, including influenza A, Ebola, hepatitis B and C, human immunodeficiency virus 1, human papillomavirus, West Nile, and Zika [37]. Nevertheless, many other viruses, such as Middle East and severe acute respiratory syndromes, foot and mouth disease virus, or influenza B, deploy mechanisms to prevent the antiviral effect of ISG15 [38,39,40,41,42]. Interestingly, in addition to its role as an effector ISG, ISG15 also prevents the over amplification of the IFN-I cascade [41]. It is then plausible that the lower expression of ISG15 in the absence of PKR at early time points upon TCRV infection disrupts the regulatory feedback loop, failing to repress the IFN-I response at later times, which may result in the increased expression of Mx1 and ISG15. Although hemorrhagic syndrome caused by OW arenavirus is not associated to a cytokine storm [43,44], the symptomatology of highly pathogenic NW arenaviruses seems to correlate with activation of infected macrophages that leads to massive release of proinflammatory cytokines [45]. Furthermore, LASV, but not highly pathogenic NW arenaviruses, prevents the accumulation of the dsRNA danger signal which leads to PKR activation [25]. Therefore, it is conceivable that PKR activation and the subsequent dysregulation of ISG15 may alter the cytokine release pattern, with consequences in the symptomatology caused by mammarenaviruses. A detailed investigation is encouraged to determine the biological and clinical consequences of ISG15 altered expression upon mammarenavirus infection.

Mx1 is an effector ISG with GTPase activity and an antiviral effect against several viral infections, including influenza, bunyaviruses, and hantaviruses [46,47,48,49]. When activated, Mx1 oligomerizes and sequesters viral factors of LaCrosse virus, influenza A, or Thogoto virus, disturbing the viral life cycle [46,49,50]. Unlike its mouse orthologues, human Mx1 locates in the cytoplasm and is effective against a broad range of viruses, regardless their replication site [47,48]. To the current date, there is no reported inhibition of mammarenaviruses by Mx1. The different consequences of PKR on Mx1, ISG15, and CCL5 expression kinetics suggest that, although all these genes are overall upregulated upon viral infections, they are also differently tuned by additional host factors such as PKR.

A complete vision of the innate immune response and its modulation is crucial to understand the mechanisms underlying the pathogenicity of mammarenaviruses, and for the development of new antiviral strategies. The results presented here contribute to a better understanding of the dynamics in the onset of the innate immune response triggered by NW arenavirus infections and, in particular, of the key role of PKR on it. The change in the kinetic and temporal expression patterns of Mx1 and ISG15, but not of CCL5, demonstrates that PKR does not only modulate the amplitude of the IFN-I response, but also tunes the expression of selected ISGs over time throughout mammarenavirus infection. These findings highlight the importance of the ability to control PKR activation during an NW arenavirus infection, and the potential consequences for a better understanding of the hemorrhagic syndromes caused by highly pathogenic mammarenaviruses.

## Figures and Tables

**Figure 1 viruses-13-01313-f001:**
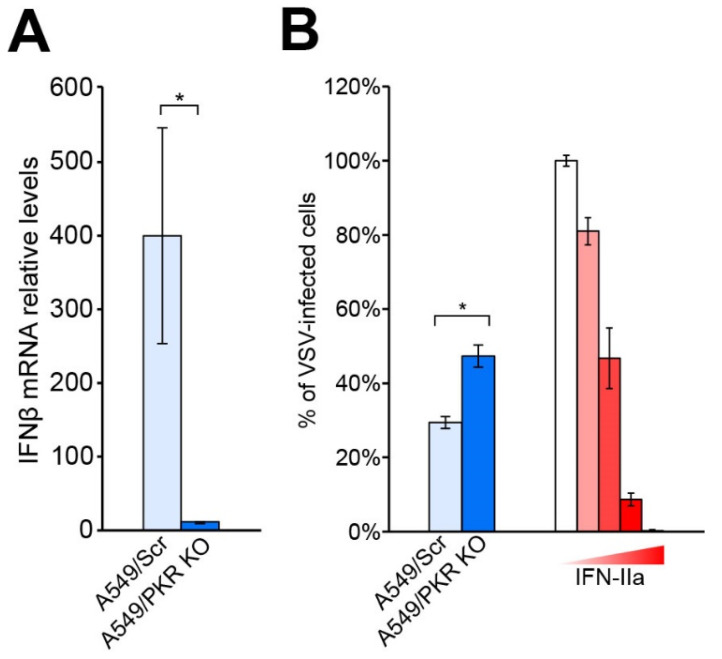
(**A**) IFN-β mRNA expression in A549/Scr and A549/PKR KO cells infected with TCRV. Cells were infected at MOI 0.01 PFU/Cell with TCRV, total RNA was collected 5 days after infection and subjected to RT-qPCR as described in Materials and Methods. Fold-induction was calculated by the 2^−^^ΔΔCt^ method. Error bars represent standard deviations (*n* = 6). (**B**) Detection of IFN activity by bioassay. Conditioned supernatants from TCRV-infected A549/Scr and A549/PKR KO cells were UV inactivated for 2 min and used to pretreat VeroE6 cells for 16 h. As a positive control, we used titrated amounts of IFN-IIa (10^4^, 10^3^, 10^2^, 10, and 0 IU/mL) diluted in supernatant from uninfected cells. Pretreated cells were infected with VSV (300 PFU/well). Values were normalized to samples in absence of IFN. Error bars represent standard deviations (*n* = 3) of results from one representative experiment out of two independent experiments. * stands for *p* value < 0.001 in an ANOVA test.

**Figure 2 viruses-13-01313-f002:**
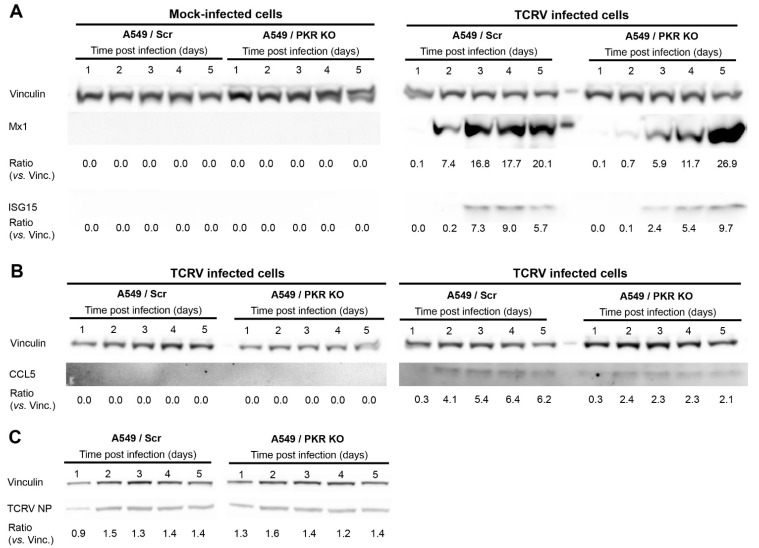
Protein expression of Mx1, ISG15, and CCL5 in TCRV-infected A549/Scr and A549/PKR cells. Cells were infected with TCRV at MOI of 0.01 PFU/cell and cell lysates were collected every 24 h. Total cellular proteins were probed for Mx1, ISG15 (**A**), and CCL5 (**B**), expression by Western blotting. Vinculin was included as a loading control. The ratios of Mx1, ISG15, and CCL5 versus Vinculin were calculated by densitometry at the corresponding day postinfection. One representative example out of three independent experiments is shown; (**C**) TCRV NP expression in A549/Scr and A549/PKR KO cells.

**Figure 3 viruses-13-01313-f003:**
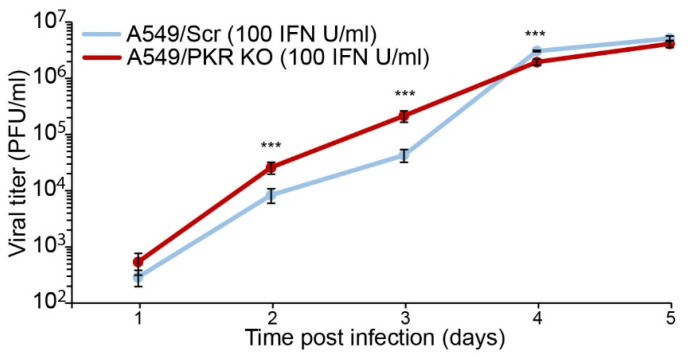
TCRV viral progeny production in IFN-stimulated A549/Scr and A549/PKR KO cells. Cells were stimulated with 100 IU/mL of rIFN-αA/D 24 h before infection. Cells were infected at MOI 0.01 PFU/cell and supernatants were assayed for viral titer by IFA. Error bars represent standard deviations (*n* = 4) and asterisks (***) means statistical significance (*p* < 0.01) in two-way ANOVA test.

**Table 1 viruses-13-01313-t001:** ISG profile of A549/Scr and A549/PKR KO cells infected with TCRV. Cells were infected at MOI of 0.01 PFU/cell and collected 5 days after infection. Values correspond to fold-induction (calculated by the 2^−^^ΔΔ^^Ct^ method) and asterisks means statistical significance when compared to uninfected cells (* *p* < 0.01, ** *p* < 0.001) in two-way ANOVA test. For clarity, color intensity is proportional to up- or down-regulation (red and blue, respectively).

ISG Name (Gene ID)	A549/Scr	A549/PKR KO	ISG Name	A549/Scr	A549/PKR KO
***CCL5*** (6352)	12,673.81 **	5135.67 **	***RELA*** (5970)	2.10	0.84
***IFNB1*** (3456)	5028.57 **	2020.63 **	***ATG5*** (9474)	1.87	1.61
***MX1*** (4599)	1046.76 **	1753.85 **	***IRF3*** (3661)	1.86	1.72
***ISG15*** (9636)	622.35 **	751.67 **	***APOBECC3G*** (60489)	1.82	1.45
***OAS2*** (4939)	470.5 **	304.32 **	***TBK1*** (29110)	1.76	2.85
***IL6***(3569)	142.55 **	32.25 **	***RIPK1*** (8737)	1.74	0.81
***IFIH1*** (64135)	56.58 **	62.8 **	***CTSL*** (1514)	1.72	2.27
***IRF7*** (3665)	35.41 **	14.54 *	***IL15*** (3600)	1.71	1.26
***DDX58*** (23586)	30.11 **	38.58 **	***IL18*** (3606)	1.70	1.31
***CXCL8*** (3576)	12.4 **	7.49 **	***CD80*** (941)	1.56	2.82
***TLR3*** (7098)	10.65 **	12.05 **	***MAP2K1*** (5604)	1.54	1.99
***IL12A*** (3592)	9.34 **	5.27 **	***CASP10*** (843)	1.52	0.89
***STAT1*** (6772)	9.25 **	8.39 **	***MAP3K7*** (6885)	1.49	1.48
***CYLD*** (1540)	5.67 **	4.23 **	***DDX3X*** (1654)	1.41	1.06
***TICAM1*** (148022)	3.7 **	1.23	***IRF5*** (3663)	1.41	1.10
***NFKB1A*** (4792)	3.49 **	2.69	***IFNAR1*** (15975)	1.37	1.05
***IRAK1*** (3654)	3.23	0.77	***MAPK1*** (5594)	1.35	1.29
***TRIM25*** (7706)	3.22 **	1.64	***IKBKB*** (3551)	1.30	0.64
***NFKB1*** (4790)	3.17 *	1.40	***MAPK8*** (5599)	1.30	1.00
***MYD88***(17874)	3.11 **	2.86 **	***CASP8*** (841)	1.26	1.08
***TRADD*** (8717)	2.94	2.53	***SPP1*** (6696)	1.21	2.07
***CD40*** (958)	2.8	1.45	***PIN1*** (5300)	1.16	1.53
***MAP2K3*** (26397)	2.78 **	1.96 **	***SUGT1*** (10910)	1.13	1.69
***AZI2***(64343)	2.73 *	2.31	***HSP90AA1*** (3320)	1.05	1.27
***TRAF6*** (7189)	2.7 **	1.78 *	***CARD9*** (64170)	1.04	0.78
***CHUK***(1147)	2.63 **	2.37	***MAPK14*** (1432)	0.97	0.67
***FADD*** (8772)	2.63 *	1.72	***MAPK3*** (5595)	0.93	0.45
***CTSS*** (1520)	2.41 **	2.61 **	***CTSB*** (1508)	0.91	1.30
***MAVS*** (57506)	2.4 **	2.26	***IL1B*** (3553)	0.85	0.83
***TRAF3*** (7187)	2.36	1.34	***PYCARD*** (29108)	0.73	1.13
***CXCL9*** (4283)	2.13	3.21	***TKFC*** (26007)	0.72	0.34
***MAP3K1*** (4214)	2.12 **	0.91	***FOS*** (14281)	0.61	0.28
***JUN*** (3725)	2.11	0.59	***IFNA1*** (3439)	0.61	0.37

## Data Availability

All relevant data were within the manuscript.

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
