# Peer review of "The Protein Kinase Receptor Modulates the Innate Immune Response against Tacaribe Virus"

_viruses, 2021, doi:10.3390/v13071313_

Round 1

Reviewer 1 Report

This paper by Moreno and Kunz presents studies aimed at characterizing the role of PKR in Tacaribe virus (TCRV) infection. The present work is a logical follow up to studies previously published by this group showing that TCVRV infection triggers a strong host cell innate immune response.  The present work compares the host cell innate immune response to TCRV in the presence and absence of PKR.

The overall experimental design is sound and the authors have presented results supporting an important role of PKR in the regulation of the host cell innate immune response to TCRV infection. The key message of the paper is that TCRV infection of PKR KO A549 cells, compared to control cells, resulted in diminished transcriptional activation of  IFNb that correlated with altered levels and temporal expression of some specific ISGs, which correlated with increased levels of production of TCRV infectious progeny at early (2 and 3 days) times of infection.

There are several aspects of the experimental section of the paper that require some improvements:

1) Figure 1 should incorporate information about the numbers of infected cells and overall viral load at the time used to determine levels of IFNb mRNA. It would be also important to provide information about the correlation of transcriptional activation of IFNb and production of bioactive IFN-I, as RNA levels do not necessarily correlated with biological activity.

2) Results shown in Figure 2 need to incorporate mock-infected controls for days 1 and 5, as well as levels of viral protein for all time points examined.

3) Results shown in Figure 3 need to incorporate viral growth curves in WT and PKR-KO A549 cells that were not stimulated with IFN-I prior infection. In addition, the results will be strengthened by the incorporation of reconstitution experiments in the PKR-KO cells to confirm that absence of PKR is the determinant of the observed phenotype.

The introduction needs some editing to correct inaccuracies:

1) Mammarenavirus is not a family but rather a genus within the family Arenaviridae.

2) Better use the term mammarenavirus referring to NW and OW arena viruses.

3) NP has a 3'-5' exonuclease rather than 5'-3' exonuclease.

4) Genome segment rather than genomic fragment.

5) Lines 51-53, needs rewriting.

Reviewer 2 Report

The manuscript by Moreno et al further refines the role of PKR in controlling Tacaribe virus (TCRV), a non-pathogenic new world arenavirus. The study uses A549 cells and those with a PKR knockout to compare the innate immune response, particularly related to type I interferon and ISGs, when infected with TCRV. Similar experiments were performed in this group's previous work (JVI 2019), and although the comparison for that paper focused on TCRV and Junin, the conclusion for this paper and their previous work report a specific role for PKR in TCRV in arenavirus infection.

Here, the authors identify a delayed response in ISG genes Mx1 and ISG15 and an overall inhibition of CCL5 in PKR KO cells compared to control cells. Additionally, they suggest PKR partially restricts TCRV growth when infecting PKR KO and control cells with TCRV after IFN stimulation. These are novel results but with a narrow focus of only one cell type.

Specific comments:

In figure 1, the error bar for TCRV infection in A549/Scr cells seems high and therefore the authors should double check the statistical significance between TCRV and JUNV in these control cells.

In figure 2, the difference between IFN-b expression in control vs KO cells is only 2.5 fold however, in figure 1  the difference is about 400 fold. The authors claim the profiling method in figure 2 is more sensitive RT-qPCR conducted in Fig1, which may be the case but that does put into question the difference between TACV and JUNV in first figure which is about 2 fold difference but might be quite different if more sensitive method were used.

In figure 3, there is a label missing as I believe the left hand panels are uninfected cells and right hand panels are infected. This should be correctly labeled.

In figure 4, error bars for day 4 are not present and should be double checked. It appears that day 4 and day 5 viral titers are very similar in KO versus control cells which means PKR only restricts growth in early time points.

Author Response

This manuscript is a resubmission of an earlier submission. The following is a list of the peer review reports and author responses from that submission.